# Effects of Microencapsulated Blend of Organic Acids and Essential Oils as a Feed Additive on Quality of Chicken Breast Meat

**DOI:** 10.3390/ani10040640

**Published:** 2020-04-07

**Authors:** Alessandro Stamilla, Nunziatina Russo, Antonino Messina, Carmine Spadaro, Antonio Natalello, Cinzia Caggia, Cinzia L. Randazzo, Massimiliano Lanza

**Affiliations:** 1Department of Agriculture, Food and Environment (Di3A), University of Catania, Via Valdisavoia, 5, 95123 Catania, Italy; alessandrostamilla@gmail.com (A.S.); nunziatinarusso83@gmail.com (N.R.); carminespad@gmail.com (C.S.); ccaggia@unict.it (C.C.); cranda@unict.it (C.L.R.); malanza@unict.it (M.L.); 2DVM consultant poultry specialists, Via Cava Gucciardo Pirato, 12, 97015 Modica, Italy; vetmessina@gmail.com

**Keywords:** essential oils, organic acids, poultry meat, feed additive, meat shelf-life, meat quality

## Abstract

**Simple Summary:**

Chicken meat is largely consumed around the world, with an increasing demand in recent years. Unfortunately, chicken meat is very susceptible to oxidative deterioration; therefore, poultry industries often use synthetic dietary additives to improve meat shelf-life. However, due to the public’s growing concern about the potential toxic effect of the synthetic additives, there is an increasing interest in natural antioxidant compounds. Among these, organic acids and essential oils could represent a favourable option to improve the characteristics of chicken meat. The aim of this study was to evaluate the dietary supplementation of organic acids (sorbic and citric) and essential oils (thymol and vanillin) on the quality and shelf-life of broiler meat. The additive was supplemented for the entire growing cycle at the level of 0.5% (as a feed additive). This dietary strategy led a reduction in intramuscular fat content and an overall improvement in fatty acid profile. Moreover, the dietary supplementation of organic acids and essential oils reduced the lipid oxidation in cooked meat, whereas minor changes were observed for colour and lipid stability and for microbial loads in raw meat.

**Abstract:**

The present study aims to investigate the effect of dietary supplementation based on a blend of microencapsulated organic acids (sorbic and citric) and essential oils (thymol and vanillin) on chicken meat quality. A total of 420 male Ross 308 chicks were randomly assigned to two dietary treatments: the control group was fed with conventional diet (CON), while the other group received the control diet supplemented with 0.5% of a microencapsulated blend of organic acids and essential oils (AVI). In breast meat samples, intramuscular fat content and saturated/polyunsaturated fatty acids ratio were reduced by AVI supplementation (*p* < 0.05). Moreover, atherogenic (*p* < 0.01) and thrombogenic (*p* < 0.05) indices were lower in AVI than CON treatment. AVI raw meat showed a lower density of psychrotrophic bacteria (*p <* 0.05) at an initial time, and higher loads of enterococci after 4 days of refrigerated storage (*p* < 0.05). No contamination of *Listeria* spp., *Campylobacter* spp., and *Clostridium* spp. was found. TBARS values of the cooked meat were lower in the AVI treatment compared to CON (*p* < 0.01). Among colour parameters, a*, b* and C* values increased between 4 and 7 days of storage in AVI cooked meat (*p* < 0.05). Overall, organic acids and essential oils could improve the quality and shelf-life of poultry meat.

## 1. Introduction

The popularity of chicken meat has been increasing around the world in recent years, and, in 2018, the consumption of poultry meat reached a value of 30.6 kg per inhabitant in the countries of the Organization for Economic Co-operation and Development [1]. It has been clearly demonstrated that the shelf-life of meat mainly depends on the storage conditions and the quality of meat, which is strictly connected to the life cycle of animals before slaughter [2]. Generally, there are many factors affecting the quality of meat and meat spoilage is often associated to physical, chemical and microbiological deterioration. Even if numerous attempts have been made, chicken meat remains a highly perishable commodity which deteriorates after 4–10 days’ post slaughter, in refrigerated conditions, with the main economic losses being due to growth of spoilage microorganisms [3,4]. Furthermore, chicken meat has been frequently found contaminated with pathogenic bacteria, such as *Salmonella* spp., *Campylobacter* spp., verocytotoxigenic *Escherichia coli* (VTEC), *Yersinia enterocolitica, Listeria monocytogenes*, for which microbiological criteria are defined both by Regulation (EC) 853/2004 and Commission Regulation (EU) 1495/2017.

In general, meat is inclined to oxidative deterioration [5] and, in particular, to lipid oxidation, which affects colour, flavour, odour, texture, and nutritional value [6,7]. The lipid oxidation is frequent in poultry due to the high content of polyunsaturated fatty acids (PUFA). Moreover, the rate of oxidation is related to the damage of tissue, stress and physical damages occurring during pre-slaughter and the rearing period, and to early post-mortem conditions, such as pH and the temperature of carcasses [8].

Although the chicken meat perishability depends, firstly, on the storage conditions, its quality is strictly connected with animal feeding. For this reason, dietary supplementation is regarded as the simplest way to affect meat composition and several studies focused on the effects of feed supplementation in improving the oxidative stability of the tissue [9,10,11] through the introduction of soluble antioxidants into phospholipid membrane tissues [12]. Butylated hydroxytoluene (BHT) has been widely used as an antioxidant, but the recent trend to shift from synthetic to natural antioxidants has allowed the scientists to test numerous compounds derived from plants (e.g., grape, rosemary extract, etc.), animals (e.g., chitosan from fish), and microorganisms (e.g., bacteriocins) [13,14].

Essential oils (EO) are mixtures of volatile compounds present as secondary compounds in plants, from which they are extracted by steam distillation or solvent extraction. They have been largely proven to exert a natural antibiotic and antioxidant action [11,15,16]. Among EO, thymol is the most studied phytochemical bioactive compound able to modify the bacterial cell membrane permeability [17] and to react with lipid and hydroxyl radicals [18]. As a feed additive, thymol has been found to be effective in prolonging the shelf-life of chicken meat [11]. Furthermore, vanillin, the most common flavouring agent, has been proven to have an antimicrobial role and modulation effects in intestinal microbiota, also improving the nutrient absorption [19]. Due to the fast metabolic conversion of EO and excretion, the accumulation of EO in poultry meat can be collected only if chickens are continuously fed with EO [20].

Lastly, organic acids (OA), as sorbic and citric acids, have been historically used as feed additives for their antimicrobial properties. A film package containing chitosan microcapsules of citric acid has showed antimicrobial and antifungal proprieties, and the addition of citric acid in meat was found to be efficient in reducing the microbial loads in poultry by 10% [21,22], reducing mesophilic and psychrotrophic bacteria [23] and, in its dissociated form, *Campylobacter jejuni* species [24].

The feed supplementation with OA and EO is well documented in swine either alone or in combination with or in partial replacement of allopathic growth promoters (antibiotics) due to their acidifying and antimicrobial properties [25,26]. A commercial microencapsulated blend of OA (namely citric and sorbic acid) and synthetic EO (namely thymol and vanillin), has already been authorized as an additive for swine and poultry, according to EU Regulation 849/2012 [27]. Its use, mainly as replacer of antibiotic growth promoters, is well documented either in swine [28] and in poultry, for which it is reported to improve feed intake, food conversion ratio, and to decrease mortality [29]. In a recent study, we observed a reduction in the overall mortality rate and a positive effect on growth rate in the last period of the growing cycle when broiler chicken diet was supplemented with 0.5% of microencapsulated blend of OA and EO [30]. Moreover, in the same study, a favourable effect on gut morphology was also found in different intestinal segments, in the last growing phases. However, little or any information is available on the potential effects of OA and EO feed supplementation on poultry meat quality.

Therefore, the aim of the present study was to determine whether the dietary supplementation of microencapsulated blends of OA and EO affects the quality and shelf-life of breast chicken fillets. To achieve this objective, the same chickens from the study by Stamilla et al. [30] were used to evaluate the oxidative stability and microbial dynamics in raw and cooked meat stored in practical conditions. Moreover, the fatty acid profile and physical parameters of the meat were determined.

## 2. Materials and Methods

### 2.1. Experimental Design and Diets

The animals were handled by specialized personnel following the European Union Guidelines (2010/63/EU Directive). The experimental design was in detail described by Stamilla et al. [30]. Briefly, 420 male Ross 308 chicks (1 day old) from the same hatching were randomly allocated in six pens, all placed in the same shed. Each pen (3.5 × 1.5 m) of 70 chicks was assigned to one of the two dietary treatments (three pens/treatment). The control group was fed with a commercial diet (CON), while the other group received the same diet supplemented with a 0.5% of microencapsulated blend of OA and EO (AVI). The microencapsulated feed additive (AviPlus^®^ P) was obtained from Vetagro S.p.A. (Reggio Emilia, Italy) and it contains organic acids such as citric (25%) and sorbic acids (16.7%) and synthetic essential oils, such as thymol (1.7%) and vanillin (1%). Chicks were reared on a comminute straw litter and each pen was equipped with nipple drinkers and plastic feeders to provide ad libitum feed intake and free access to water. Animals were fed across four different growing phases, with a changing basal diet composition: starter (0–12 days), grower 1 (12–26 days), grower 2 (26–35 days) and finisher (35–47 days), as reported in Table 1. The composition of the nutrients in each basal diet was planned to satisfy the nutritional requirements of chicks, according to National Research Council (NRC) [31].

The chicks were vaccinated against Infectious Bursal Disease Virus (IBVD), Infectious Bronchitis (IB, 793b, H120) and Marek’s disease. The feed used for the whole experimental trial contained a coccidiostat, Nicarbazin 40 ppm and Narasin 50 ppm. The applied light program was the following: 0–24 days (20 h light/4h dark), 25–33 days (23 h light/1h dark), 34–38 days (22 h light /2h dark), 39–47 days (23 h light/1h dark). The shed microclimatic conditions were taken under a strict control, according to general guidelines for poultry production. The temperature ranged from 32 to 40 °C for the first week; it was subsequently reduced to about 3 °C a week, up when to the chicks were slaughtered; the humidity was kept at a constant value, and adequate ventilation was guaranteed, through screened windows with mosquito nets and sometimes with a fan.

### 2.2. Slaughter Procedures and Meat Sampling

On the 47th day of the trial, after a 12 h period of fasting, the broilers were transferred to the slaughtering house, stunned with carbon dioxide and processed according to the Council Regulation EC no. 1099/2009 on animal welfare at slaughter and to the CE Regulation n. 853/2004 on the hygiene of foodstuffs. Carcasses were automatically processed (plucked, separated by head and legs and eviscerated) and transferred into a tunnel at 4 °C for 3 h. Carcasses were weighted and sectioned to calculate the percentage of yield at slaughter and the incidence (as percentage) of breast, thigh and wings.

At the end of the slaughtering process, four entire chicken breasts (pectoral major muscle) for each replicate (pen) were randomly selected (12 CON and 12 AVI), individually packaged in oxygen-impermeable vacuum bags and immediately transferred into the laboratory in refrigerated conditions. Then, each muscle sample was split into seven aliquots: two portions (about 90 g) were used for the evaluation of drip loss and cooking loss; one portion was vacuum packed and stored at −20 °C for fatty acid determination; two aliquots were arranged for microbiological shelf-life analyses (raw and cooked); and the remaining two aliquots were used to assess the oxidative stability of raw and cooked meat.

### 2.3. Analyses of Feeds

Two samples (CON and AVI) for each feed period, for a total of eight CON and eight AVI diet samples, were collected during the trial, vacuum packed and stored at −20 °C until analyses were done. Dry matter (DM), crude protein, lipid, crude fibre, ash, calcium, phosphorus, lysine and methionine were detected according to Commission Regulation (EC) N. 152/2009 [32], which fixed the sampling procedures and analytical methods for the official control of feed. Total phenols and total tannins were extracted from 200 mg of feed samples using 10 mL acetone 70% (*v/v*) and quantified by the Folin-Ciocalteu method, as detailed in Valenti et al. [33]. Fatty acids methyl esters (FAME) of feeds were prepared by a one-step procedure using chloroform and 2% (*v/v*) sulfuric acid in methanol [34]. Nonadecanoic acid methyl ester (C19:0) was used as internal standard, and gas chromatographic analysis was applied as described below for intramuscular fatty acid composition.

### 2.4. Determination of pH, Drip and Cooking Losses

Drip loss was determined on meat samples after 72 h of storage at 4 °C and expressed as a percentage of the difference between weights, according to Marcinkowska-Lesiak et al. [35]. Meat samples were cooked via the direct immersion of the vacuum bag in a water bath at 70 °C for 30 minutes. Cooking loss was determined right after the cooking step according to Alves et al. [36]. Meat pH was measured in duplicate by a pH meter (HI-110; Hanna Instruments, Padova, Italy) from the whole meat samples after 24 h post-mortem or immediately after cooking.

### 2.5. Intramuscular Fatty Acid Composition

The intramuscular fat (IMF) was extracted from 10 g of finely minced *pectoralis major* samples with a mixture of methanol and chloroform (1:2, *v/v*). An amount of 50 mg of extracted lipids was methylated by base-catalysed transesterification, using 1 mL of sodium methoxide in methanol 0.5 N and 2 mL of hexane containing nonadecanoic acid as an internal standard [34]. FAME were analysed using a Thermo Finnigan Trace gas-chromatograph (ThermoQuest, Milan, Italy), equipped with a flame ionization detector and a high-polar capillary column (SP-2560 fused silica, Supelco, Bellafonte, PA, 100 m × 0.25 mm i.d.; film thickness 0.25 μm). Chromatographic conditions were as reported by Natalello et al. [37] and fatty acids were expressed as mg/100 g of meat. Atherogenic index (AI) and the Thrombogenic index (TI) were also calculated according to Ulbricht and Southgate [38], in order to evaluate the risk of atherosclerosis and the potential aggregation of blood platelets, respectively.

### 2.6. Meat Shelf-Life

With the aim of evaluating the shelf-life of raw and cooked meat samples, conventional storage conditions were simulated. Breast samples were split into four aliquots and stored at 4 °C on a plastic tray covered by a domestic cling film, both in polyethylene. The shelf-life of meat samples was evaluated following microbial dynamics, colour parameters and the lipid oxidation of both raw and cooked samples at 0, 4, 7, 11 days and at 0, 2, 4, 7 days of storage, respectively.

For microbiological analyses, 25 g of samples were appropriately diluted in physiological saline solution, homogenized by a stomacher and subjected to conventional counting methods. The media and conditions used were as follows: Violet Red Bile Glucose Agar (VRBGA), aerobically incubated at 37 °C for 24 h and at 45 °C for 48 h, for *Enterobacteriaceae* and faecal coliforms counts, respectively; chromogenic E. coli, incubated at 37 °C for 24 h, for *Escherichia coli* determination; Kanamicin Aesculine Azide Agar (KAA) incubated at 37 °C for 24–48 h, for enterococci determination; Plate Count Agar (PCA) incubated at 32 °C for 48 h and at 4 °C for 7 days, for mesophilic and psychrotrophic bacteria determination, respectively; Mannitol Salt Agar (MSA), incubated at 37 °C for 24–48 h, for staphylococci determination; De Man Rogosa and Sharp (MRS) agar, incubated at 37 °C for 24–48 h, for Lactic Acid Bacteria (LAB) determination; M17 agar, incubated at 37 °C for 24–48 h for lactococci count; Sabouraud Dextrose agar, incubated at 25 °C for 24–48 h for yeasts/moulds determination; Sulfite Polymyxin Sulfadiazine (SPS) agar in a double layer, incubated at 35 °C for 24–36 h for the selective enumeration of *Clostridium perfringens*. To determine the presence of *Campylobacter* spp., the following protocol was applied: 10 g of the sample was subjected to a pre-enrichment into 90 mL of Bolton broth, supplemented with horse lacquered blood (1:200), homogenized by stomacher and incubated at 40 °C for 24 h. Then, 10 µL of suspension was directly poured on modified Charcoal Cefoperazone Deoxycholate Agar (mCCDA) base [39]. The presence of *Listeria monocytogenes* was detected as follows: 10 g of the sample was put into 90 mL of Half Fraser broth and homogenized by a stomacher before incubating at 30 °C for 24 h. Then, 1 mL of suspension was poured into a tube previously filled up with Fraser Broth and incubated at 30 °C for 24 h. The suspension was poured on Aloa Agar and plates were incubated at 30 °C for 24 h [40]. Lastly, the detection of *Salmonella* spp. was carried out as follows: 10 g of sample was added to 90 mL of Tryptone Soya Broth, homogenized by a stomacher and incubated at 37 °C for 24 h. Then, 1 mL of suspension was poured into Modified Semi-Solid Rappaport–Vassiliadis agar medium, and plates were incubated at 40 °C for 48 h [41]. Results were expressed as the log_10_ colony forming unit per gram of the sample (log_10_ cfu/g).

The colour measurement, along with the storage time, of both raw and cooked meat samples (1.75 cm-thickness) was carried out using a Minolta CM-2022 spectrophotometer (d/8° geometry; Minolta Co., Ltd. Osaka, Japan) in the CIE L*a*b* space, using illuminant A and 10° standard observer. Lightness (L*), redness (a*), yellowness (b*), chroma (C*), hue angle (H*) and the reflectance spectra, between 400 and 700 nm, were measured for each sample. To obtain an average value for all the measured parameters, two readings were registered. The ratio (K/S)_572_ ÷ (K/S)_525_ was also calculated to detect the accumulation of metmyoglobin on both the raw and cooked meat surface at each storage time [42]. Overall colour variation (ΔE) between each storage time and initial time was calculated as follows: ΔE = [(ΔL*)^2^ + (Δa*)^2^ + (Δb*)^2^]^1/2^, where ΔL*, Δa* and Δb* were the differences between L*, a* and b* values at each sampling time and their values detected at initial time (day 0), whereas the whiteness index (WI) was calculated as follows: WI = 100 – [(100 − L*)^2^ + (a*)^2^ + (b*)^2^ ]^1/2^.

Lipid oxidation of meat samples was determined by detection of 2-Thiobarbituric acid-reactive substances (TBARS) according to Sinnhuber and Yu [43] with some modifications. Briefly, 1 g of meat was homogenized for 1 min (DIAX 900, Heidolph, Kelheim, Germany) with 3 mL of 1% thiobarbituric acid (TBA) solution (0.02 mM) and 17 mL of 25% trichloroacetic acid (TCA). The homogenized sample was heated in a boiling water bath for 30 min and then cooled in cold water for 10 min. Then, 5 mL of supernatant was collected, mixed with 3 mL of chloroform and centrifuged at 3200 × *g* for 30 min. The absorbance of samples was measured at 532 nm using a Shimadzu UV-vis spectrophotometer (UV-1601; Shimadzu Corporation, Milan, Italy). The assay was calibrated with a solution of known concentration of 1.1.3.3-tetraethoxypropane (TEP) in distilled water [44]. Results were expressed as mg of malondialdehyde (MDA)/kg of meat.

### 2.7. Statistical Analyses

In order to test the effect of dietary treatment (CON vs. AVI), data on pH, drip loss, cooking loss and fatty acid composition were analysed using a one-way ANOVA. Microbiological and oxidative stability data were analysed for repeated measures using a mixed model to test the effect of dietary treatment and storage time, as well as the effect of their interaction as fixed factors, while each chicken was considered as random factor. Differences between means were assessed using Tukey’s adjustment for multiple comparisons. Significance was declared when *p* ≤ 0.05, while trends were considered for 0.05 < *p* < 0.10. Statistical analyses were performed using the statistical software Minitab, version 16 (Minitab Inc, State College, PA, USA).

## 3. Results

### 3.1. Slaughter Performances and Physical Characteristics of Meat

Data on slaughter performances, drip loss at 72 h, cooking loss and pH in raw and cooked meat are reported in Table 2. No significant differences between treatments were found for carcass yield, breast, thigh and wings percentages. Similarly, no significant variations between treatments were observed for all the above physical parameters of meat.

### 3.2. Intramuscular Fatty Acid Composition

The intramuscular fat (IMF) content and the fatty acid composition of the breast meat is reported in Table 3. The IMF values were significantly affected by dietary treatment (*p* < 0.05) and were lower in AVI meat samples compared to the counterpart. The AVI feeding diet significantly (*p* < 0.05) lowered the saturated/polyunsaturated fatty acids ratio in meat compared to the CON dietary treatment. Meat obtained from CON chickens showed a greater proportion of myristic acid (C14:0; *p* < 0.05), C17:0 iso (*p* < 0.05), trans-palmitoleic acid (C16:1 trans-9; *p* < 0.01), C18:1 trans-10 (*p* < 0.05), C20:0 (*p* < 0.01), C20:5 n-3 (*p* < 0.01), and in tendency of pentadecanoic acid (C15:0; P < 0.10) and margaric acid (C17:0; *p* < 0.10) compared to AVI meat. Conversely, AVI meat samples showed higher proportion of C18:2 trans-9, trans-12 (*p* < 0.05) and docosadienoate (C:22:2 n-6; *p* < 0.05). Both atherogenic (*p* = 0.010) and thrombogenic (*p* = 0.013) indices were significantly affected by dietary treatments, resulting in lower values in AVI meat compared to CON samples.

### 3.3. Microbiological Results

In the present study, the main microbial groups were detected in raw and cooked meat samples, during storage time (at 0, 4, 7, and 11 days, in raw meat, and at 0, 2, 4, and 7 days, in cooked meat) at refrigerated conditions. The effect of dietary treatment, storage time and their interaction on microbial counts (expressed as log_10_ cfu/g) in AVI and CON samples during storage at refrigerated conditions are reported in Table 4 and Figure 1. Overall, *Listeria* spp., *Campylobacter* spp., and *Clostridium* spp. were never found during the storage period. No significant effects of dietary treatment were found for the detected microbial groups. Looking at results obtained at initial times, the highest mean microbial densities were detected for mesophilic bacteria (4.2 log_10_ cfu/g), *Enterobacteriaceae* (4.0 log_10_ cfu/g) and yeasts/moulds (3.8 log_10_ cfu/g), whereas the lowest values were detected for lactococci (1.3 log_10_ cfu/g) and enterococci (1.4 log_10_ cfu/g). In general, a significant increase in all detected microbial groups was revealed during storage, except for *E. coli*, for which the increases were never significant.

Mesophilic bacteria, *Enterobacteriaceae* and yeast/mould loads reached values of about 8 log_10_ cfu/g after 7 days and values higher than 9.47 log_10_ cfu/g at the eleventh day of refrigerated storage, both in AVI and CON samples (Appendix A). The interaction dietary treatment per storage time was significant for *Enterococcus* spp. (*p* < 0.001), mesophilic bacteria (*p* < 0.05), psychrotrophic bacteria (*p* < 0.01), coagulase-positive staphylococci (*p* < 0.05), LAB (*p* < 0.05) and yeasts/moulds (*p* < 0.05). Comparing results between the AVI and CON meat samples within the day of storage, at initial time, no significant differences were detected among the main microbial groups, with the exception of psychrotrophic bacteria, that in AVI samples, showed significant lower (*p* < 0.05) values compared to CON samples (1.86 vs. 3.34 log_10_ cfu/g; Appendix A). After 4 days of refrigerated storage, although differences in mean microbial counts were detected among samples of the two tested trials, only enterococci showed significant (*p* < 0.05) higher values in AVI meat, compared to CON samples (5.81 vs. 2.86 log_10_ cfu/g; Appendix A). It is interesting to underscore that up to the fourth day of storage, microbial densities were basically lower in CON samples, compared to those detected in AVI samples, whereas after 7 days of storage, a turn in the trend of microbial growth was observed, with microbial counts showing lower values in AVI samples, except for lactococci, which showed significantly (P < 0.05) lower values in CON samples (5.5 vs. 6.2 log_10_ cfu/g; Appendix A). At the end of storage (eleventh, day), microbial counts were significantly lower (*p* < 0.05) in AVI samples for enterococci (6.6 vs. 7.2 log_10_ cfu/g; Appendix A) compared to CON samples. It is interesting to highlight that, in our study, the mean counts of mesophilic bacteria were always more than 1 unit log higher than psychrotrophic bacteria, with a significant difference at the eleventh day of storage. Looking at the microbial dynamics during storage, the higher differences among samples of the two dietary treatments were detected between the fourth and the seventh day of storage, with enterococci, yeasts/moulds and LAB counts that decreased mostly in AVI samples. Regarding cooked meat samples, microbiological results were always below the detectability value for all microbial groups.

### 3.4. Meat Oxidative Stability

In Table 5, the data on lipid and colour oxidative stability were reported. In detail, in the raw meat, no significant effect of dietary treatment was found on lipid oxidation (TBARS values; Figure 2a). Among colour parameters, the a* (redness) value tended (*p* = 0.075) to be higher in AVI meat compared to CON samples, while the remaining colour coordinates were not affected by dietary treatment. The (K/S)_572_ ÷ (K/S)_525_ ratio strongly tended (*p* = 0.051) to be lower in AVI meat compared to CON samples. Conversely, storage time significantly influenced all parameters (*p* < 0.001). In detail, TBARS and ΔE values increased along the storage time, while L* (lightness), a* (redness), and b* (yellowness) values dramatically increased after 11 days at refrigerated conditions, as well as C* (chroma). H* (hue angle) showed an increasing trend up to the fourth day of storage, highlighting thereafter a constant value.

In cooked meat, AVI dietary treatment decreased TBARS compared to CON (*p* < 0.01), while no significant effects were found on colour parameters. Similar to raw meat, storage time affected meat oxidative stability, showing increasing values during storage time for TBARS, b* and C* values (*p* < 0.001). L* and WI decreased until the second day of storage to remain statistically constant until the 7th day (*p* < 0.001). A similar trend was observed for a* (*p* < 0.001). Looking at H*, it was the only colour coordinate that did not change across the storage period. Comparing results between the AVI and CON meat samples within day of storage, TBARS values were lower at the fourth day of storage in AVI meat compared to CON samples (*p* < 0.05), while comparable values were detected between the two treatments for the other sampling times (Figure 2b).

Significant interactions between diet and storage time were found for a*, b* and C* in cooked meat (*p* < 0.05; Figure 3). In AVI meat, a* (redness) and b* (yellowness) indices increased between the fourth and seventh days of storage, while they did not show any variation in CON meat. As a consequence, C* chroma showed the same trend.

## 4. Discussion

Few studies have focused on the use of a blend of OA and EO in poultry, and results are often controversial compared to medicated dietary treatments [28,29]. Microencapsulation of a blend of the two groups of additives avoids the action of the stomach allowing the slow release of compounds at gut level, thus improving the additives’ antibacterial action beyond the enhancement of growth performance [45]. The present study was designed to evaluate the effects of dietary supplementation of a blend of OA (sorbic and citric acids) and EO (thymol and vanillin), already recognized as improving growth performance [28], in order to highlight the potential effects on quality of meat during refrigerated storage. In our previous trial, we observed certain positive effects on mortality rate, growth performance, gut morphology, inflammation of gut epithelium, with a reduction in *Clostridium perfringens* load in ileal content and a decrease in mesophilic bacteria and enterococci loads in litter, occurring in the last growing phase [30]. However, this positive finding did not lead to improvements in slaughter performances, which appeared comparable between treatments in terms of carcass yield and cut percentage, even if a higher yield percentage was numerically detected in AVI than in CON carcasses.

The ultimate pH values of raw meat were comparable between dietary treatments, revealing normal values for poultry meat over the threshold (≤5.91) for pale samples and under the threshold (≥6.36) for dark poultry samples, according to Swatland [46]. Accordingly, no significant differences were reported for the pH of cooked meat. Drip loss at 72 h and cooking loss were also unaffected by dietary treatments. Gheisar et al. [28] reported a linear decrease in drip loss across increasing levels of the same supplement after 5 days and up to 7 days of storage. Overall, in the present study, after 72 h of refrigerated storage, drip loss was lower than those reported (around 9.4%) by the latter authors. Poultry meat tends to lose less moisture, and for this reason it is important to measure the drip loss in time intervals longer than 24 h [47].

According to several reports, phytochemicals can improve oxidative stability of poultry meat, as by direct adding into meat or as dietary supplements [9,48,49]. Thymol seems to block the radical chain process by its interaction with peroxide radicals [50] and, recently, as a feed additive, has been found to retard lipid oxidation up to 10 days of storage, with an effectiveness comparable to butylated hydroxytoluene (BHT) [11,49]. Vanillin is mostly added, together with other compounds, in packaging formulation and its main effect is related to the antibacterial activity [51,52], especially against *E. coli* or *Campylobacter* spp.

In the present study, results on pathogenic bacteria confirmed that the applied slaughter procedures were effective in reducing contamination of carcasses from animal microbiota, such as *Salmonella* and *Campylobacter*, the two main pathogens associated with poultry meat. Furthermore, the low initial values of mesophilic bacteria highlighted a good quality of meat, corroborating that the applied air chilling process is effective in reducing growth of this microbial group. Mesophilic bacteria load is often considered as an indicator of shelf-life in poultry meat stored in air conditions [53], and in the present study the load reached the upper microbiological limit for good quality (7 log_10_ cfu/g, as defined by the ICMSF, [54]) after 4 days of storage. This result is consistent with previous findings reported for fresh poultry meat where, under refrigerated conditions, mesophilic bacteria reached that threshold within 4 or 10 days, depending on the packaging systems [53]. Regarding the *Enterobacteriaceae*, a group of Gram-negative bacteria considered a hygiene indicator during poultry processing [55], they reached the threshold (7 log_10_ cfu/g) for acceptability of chicken meat after 4 days of storage. Although LAB are considered the major microorganisms originating from the gastrointestinal tract of the broilers [56,57], in the present study, their densities, at initial times, were lower than those reported in several reports [4,48,53] and, in any case, were below 6 log_10_ cfu/g after 11 days of refrigerated storage. The low LAB and lactococci densities, together with the high densities of yeasts/moulds, could be related to the effects of both chilling procedure and packaging conditions here applied. Although low temperatures are considered relevant in delaying the *Enterobacteriaceae* and mesophilic bacteria growth [4], in the present study, their mean counts were always higher than those detected for psychrotrophic bacteria. Nevertheless, initial differences in detected microbial groups disappeared at the end of storage, as reported by Gratta and co-workers [53].

In raw meat, dietary supplementation with OA and EO tended to increase the redness a* value, compared to CON diet. Generally, in red meat, storage in overwrap packs led to the oxidation of oxymyoglobin into metmyoglobin with a decrease in the red value [58]. In chicken breast held in air at 5 °C, myoglobin and metmyoglobin were reported as the predominant pigments, whereas a lack of oxymyoglobin was observed [59]; thus, a little change in a* redness is expected through the shift from myoglobin to metmyoglobin [60]. Consistent with these results, the (K/S)_572_ ÷ (K/S)_525_ ratio tended to decrease in AVI meat compared to CON meat, indicating a potential increased level of metmyoglobin, although these differences were numerically very small and probably not perceivable by consumers. A clearer effect of AVI treatment on meat oxidative stability was shown in cooked meat in terms of lipid oxidation, with lower TBARS values in AVI compared to CON meat. Dietary supplementation mostly protected meat from lipid oxidation after 4 days of refrigerated storage, while no effect on raw meat was observed. Cooked meat is more prone to lipid peroxidation than raw meat under refrigerated storage, depending on the acceleration of oxidative reactions with lipids [61]. Regardless of the dietary treatment, TBARS values in cooked meat were much higher than those detected in raw meat. It is well known that oxidative stability in meat is regulated by the balance between pro-oxidants and antioxidants. PUFA, in particular long chain PUFA (LC-PUFA), are more susceptible to oxidation and therefore require more protective actions by antioxidants. AVI meat showed a lower SFA/PUFA ratio, which could have driven the potential increasing susceptibility to oxidation. However, dietary supplementation of OA and EO could have exerted a more efficient protective action against lipid peroxidation, thus justifying the lower TBARS values compared to the CON group. Concerning the healthiness of meat linked to dietary treatments, meat from chicken supplemented with OA and EO showed less intramuscular fat and more favourable atherogenic (AI) and thrombogenic (TI) indices. Indeed, AVI meat showed a mean AI value under the recommended threshold (< 0.5) for human health [38], in contrast to values detected in CON meat. Similarly, TI favourably decreased in AVI meat, confirming a certain favourable effect of dietary supplementation of OA and EO on the overall meat healthiness.

## 5. Conclusions

On the basis of our findings, organic acids (citric and sorbic) and essential oils (thymol and vanillin) as additives in poultry feeding could represent a valid dietary strategy to improve the quality of poultry meat. In particular, feed additive supplementation improved the intramuscular fatty acid profile, reducing atherogenic and thrombogenic indices and the lipid oxidative stability in cooked meat, while few changes in meat colour were found.

Overall, dietary supplementation did not strongly affect the microbial dynamics of breast chicken meat fillets during shelf-life at refrigerated conditions, probably due to the post slaughter chilling processing of poultry carcasses that minimized the initial bacterial load. Nevertheless, differences in microbial loads were mainly detected between 4 and 7 days of refrigerated storage. Mesophilic bacteria and *Enterobacteriaceae* loads, indicators of shelf-life and hygiene during poultry processing, respectively, reached the upper threshold for a good quality after 4 days of refrigerated storage. Both counts were not delayed by the refrigerated conditions, and were always higher than psychrotrophic bacteria loads. Low densities of LAB and lactococci, together with high levels of yeast/mould counts, were revealed and could be related to the applied packaging conditions. More studies are needed to better understand how such dietary strategies could impact on different packaging and storage temperature conditions.

## Figures and Tables

**Figure 1 animals-10-00640-f001:**
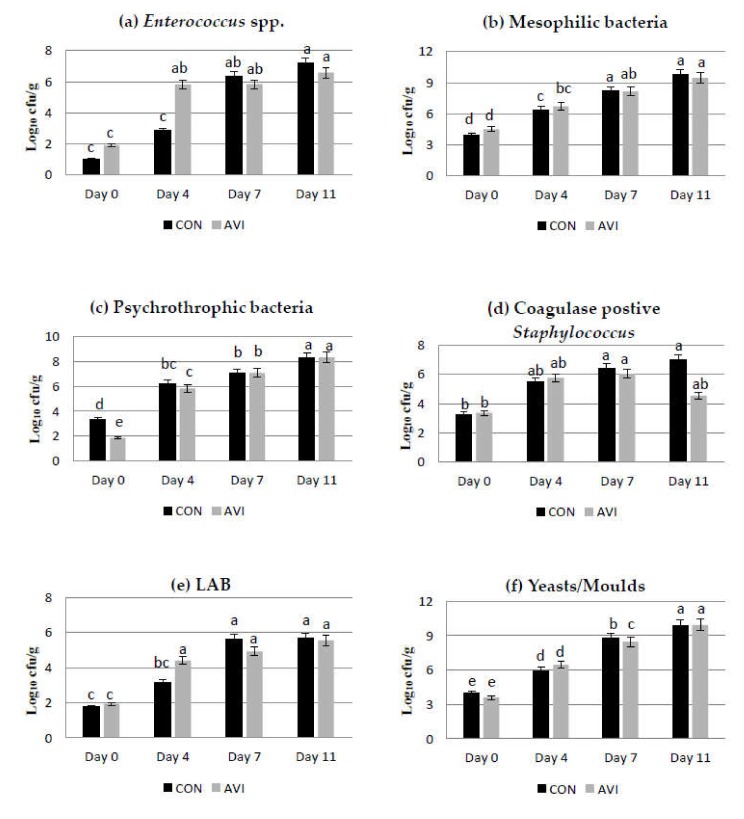
Interactive effect of dietary treatments (CON, basal diet; AVI, basal diet + 0.5% organic acids and essential oils) and storage time (days 0, 4, 7 and 11) on microbial counts of raw meat, expressed as log_10_ cfu/g: (**a**) *Enterococcus* spp.; (**b**) mesophilic bacteria; (**c**) psychrotrophic bacteria; (**d**) coagulase-positive *Staphylococcus*; (**e**) Lactic Acid Bacteria (LAB); (**f**) yeasts/moulds. ^a–e^ Values with different superscripts are significantly different (*p* < 0.05).

**Figure 2 animals-10-00640-f002:**
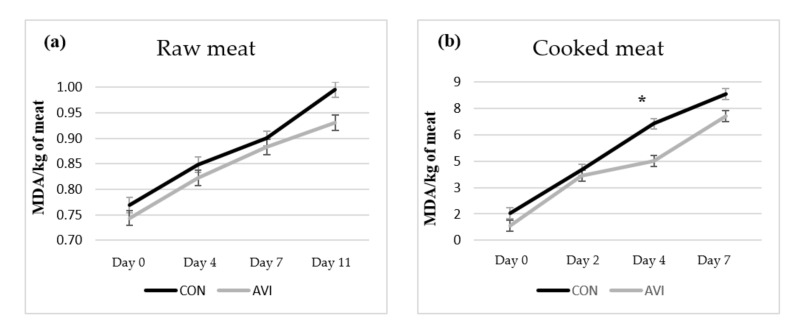
Effect of dietary treatment (CON, basal diet; AVI, basal diet + 0.5% organic acids and essential oils) on lipid oxidation trend in raw (**a**) and cooked (**b**) meat (*****: *p* < 0.05).

**Figure 3 animals-10-00640-f003:**
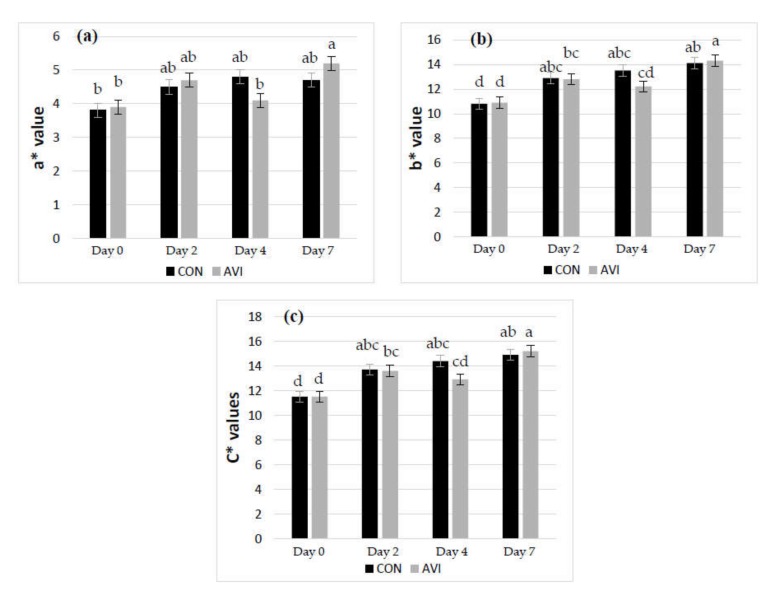
Interactive effect of dietary treatment (CON, basal diet; AVI, basal diet + 0.5% organic acids and essential oils) and time of storage (days 0, 2, 4 and 7) on a*, b* and C* values in cooked meat, (**a**), (**b**) and (**c**) respectively. ^a–d^ Values with different superscripts are significantly different (*p* < 0.05).

**Table 1 animals-10-00640-t001:** Ingredients and chemical composition of the experimental diets.

Item	Diet			
	Starter (0–12 days)	Grower 1 (12–26 days)	Grower 2 (26–35 days)	Finisher (35–47days)
Ingredients, g/100g as fed				
Corn	35	50	51	50
Soybean meal 48%	27.2	28.9	26	23.5
Soybean	10	3	2	2
Wheat	10	0	0	0
Wheat pollard	9	9	10	15
Animal Fat	3.9	4.5	6.4	5.3
Dicalcium Phosphate	1.8	1.5	1.5	1.2
Mineral premix 1	0.6	0.6	0.6	0.6
Vitamin premix 2	0.1	0.1	0.1	0.1
Calcium carbonate	0.7	0.6	0.6	0.5
Phosphate dicalcium	11.1	11.3	10.7	10.5
Chemical composition, g/100g DM				
Dry matter (DM), g/100g as fed	88.9	88.6	89.3	90.1
Crude protein	21.5	19.7	18.8	18.5
Lipid	8.99	7.23	7.75	7.86
Crude fibre	3.65	3.07	3.38	3.35
Ash	5.59	5.61	5.82	5.34
Calcium	0.87	0.84	0.78	0.63
Sodium	0.18	0.16	0.17	0.17
Phosphorus	0.61	0.61	0.59	0.55
Lysine, Lys	1.37	1.44	1.42	1.29
Methionine, Met	0.75	0.76	0.70	0.58
Metabolizable energy (kcal/kg)	3201	3060	3062	3060
Fatty acids, g/100g of total FA				
C14:0	1.41	1.08	1.25	1.02
C16:0	18.8	19.1	18.8	17.3
C16:1 *cis-*9	1.42	1.72	1.55	1.59
C18:0	9.29	9.12	9.38	7.89
C18:1 *cis-*9	25.9	26.6	26.9	25.0
C18:2 *cis-*9 *cis-*12	26.8	25.8	24.3	22.6
Phenolic compounds 3, g/kg DM				
Total Phenols	8.85	7.39	6.56	7.2
Total Tannins	6.14	4.13	3.77	4.66

^1^ Provided per kg of premix: copper (9.60 mg), iodine (0.60 mg), iron (60 mg), manganese (84 mg), molybdenum (2.4 mg), selenium (0.24 mg), zinc (84 mg), amino acids (3520 mg), sennic proteasi (15.000 PROT), enzymes (2000 PPU); ^2^ Provided per kg of premix: vitamin A (10.000 UI), vitamin D3 (3.000 UI), biotin (0.12 mg), colin (150 mg), vitamin E (36 mg). ^3^ Expressed as tannic acid equivalents.

**Table 2 animals-10-00640-t002:** Effect of dietary treatment on slaughter performances, drip loss, cooking loss and pH of raw and cooked meat.

Item	Dietary Treatment ^1^		SEM ^2^	*p*-Value ^3^
	CON	AVI		
Carcass yield (%)	67.70	67.45	0.824	0.898
Breast yield (%)	34.49	33.60	0.489	0.420
Thigh yield (%)	43.37	44.13	0.333	0.299
Wings yield (%)	18.93	18.95	0.137	0.962
pH of raw meat (at 24 h)	5.93	6.02	0.029	0.118
pH of cooked meat	6.03	6.023	0.024	0.892
Drip loss at 72 h (%)	4.97	5.41	0.468	0.650
Cooking loss (%)	28.0	27.3	0.741	0.660

^1^ CON, basal diet; AVI, basal diet + 0.5% organic acids and essential oils. ^2^ SEM, standard error of the means. ^3^
*p*-values associated with dietary treatment.

**Table 3 animals-10-00640-t003:** Effect of dietary treatment on chicken breast fatty acids composition (mg/100g of meat).

Item	Dietary Treatment ^1^		SEM ^2^	*p*-Value
	CON	AVI		
IMF ^3^ (g/100g of muscle)	2.45	1.79	0.158	0.034
C10:0	0.13	0.17	0.032	0.539
C12:0	0.62	0.45	0.065	0.205
C14:0	12.8	7.89	1.257	0.046
C14:1 cis-9	1.84	1.17	0.194	0.088
C15:0	2.40	1.54	0.230	0.062
C16:0	268	191	24.98	0.127
C17:0 iso	1.10	0.62	0.104	0.018
C16:1 trans-9	0.42	0.23	0.037	0.008
C17:0 anteiso	2.92	3.61	0.555	0.546
C16:1 cis-9	40.6	28.1	4.100	0.131
C17:0	4.03	2.59	0.388	0.061
C18:0	113	83.4	9.799	0.141
Ʃ C18:1 trans-6, 7, 8	0.14	0.09	0.021	0.219
C18:1 trans-9	2.10	1.65	0.282	0.436
C18:1 trans-10	1.81	0.90	0.201	0.020
C18:1 trans-11	3.00	2.23	0.317	0.228
C18:1 cis-6	1.26	1.07	0.173	0.601
C18:1 cis-9	210	277	51.10	0.525
C18:1 cis-11	79.3	39.4	16.01	0.220
C18:2 trans-9 trans-12	0.03	0.13	0.025	0.038
C18:2 cis-9 cis-12	241	182	23.86	0.229
C20:0	1.89	0.68	0.211	0.002
C18:3 cis-6 cis-9 cis-12	1.25	0.94	0.137	0.265
C20:1 cis-11	4.55	3.24	0.447	0.145
C18:3 cis-9 cis-12 cis-15	13.9	10.3	1.539	0.254
C20:2 cis-11 cis-14	3.93	3.19	0.307	0.233
C22:0	0.06	0.10	0.027	0.467
C20:3 n-6	4.55	3.20	0.321	0.310
C22:1 cis-13	0.25	0.26	0.029	0.944
C20:3 n-3	0.33	0.33	0.055	0.989
C20:4 n-6	24.6	20.5	1.505	0.182
C22:2 n-6	0.02	0.11	0.024	0.048
C20:5 n-3	1.59	0.88	0.124	0.002
C22:4 n-6	6.23	5.22	0.447	0.269
C22:5 n-6	0.78	0.75	0.070	0.792
C22:5 n-3	5.41	4.45	0.372	0.208
C22:6 n-3	7.01	5.57	0.521	0.174
Ʃ SFA ^4^	396	284	36.18	0.124
Ʃ MUFA ^5^	352	360	59.14	0.945
Ʃ PUFA ^6^	306	235	27.71	0.203
Ʃ PUFA n-3	28.2	21.5	2.281	0.147
Ʃ PUFA n-6	278	213	25.51	0.209
SFA/PUFA	1.28	1.20	0.018	0.028
n-6/ n-3	9.80	9.58	0.261	0.687
AI ^7^	0.57	0.40	0.035	0.010
TI ^8^	1.10	0.84	0.055	0.013

^1^ Control (CON) basal diet; basal diet + 0.5% organic acids and essential oils (AVI). ^2^ Standard error of the means (SEM). ^3^ Intramuscular fat; ^4^ Saturated fatty acids; ^5^ Monounsaturated fatty acids; ^6^ Polyunsaturated fatty acids; ^7^ Atherogenic index (AI) = (C12:0 + 4 ∗ C14:0+C16:0)/(MUFA + PUFA n−6 + PUFA n−3); ^8^ Thrombogenic index (TI) = (C14:0 + C16:0 + C18:0)/[(0.5 ∗ C18:1 cis-9) + (0.5 ∗ other MUFA)+(0.5 ∗ PUFA n−6) + (3 ∗ PUFA n−3) + (PUFA n−3/PUFA n−6)].

**Table 4 animals-10-00640-t004:** Effect of dietary treatment on microbial counts (expressed as log_10_ cfu/g) in raw meat during storage at refrigerated conditions.

Microbial Groups	Dietary Treatment (D) ^1^		Storage Time (T)				SEM ^2^	*p*-Value ^3^		
	CON	AVI	0	4	7	11		D	T	D x T
*Enterobacteriaceae*	6.940	7.191	4.008 ^d^	6.582 ^c^	8.013 ^b^	9.657 ^a^	0.316	0.416	<0.001	0.521
*Escherichia coli*	2.019	1.641	1.451	2.379	2.654	n.d.	0.295	0.789	0.017 ^4^	0.385
Faecal coliforms	3.539	3.892	2.045 ^b^	2.982 ^b^	2.595 ^b^	7.238 ^a^	0.383	0.761	<0.001	0.718
*Enterococcus* spp.	4.351	5.027	1.449 ^c^	4.335 ^b^	6.088 ^a^	6.881 ^a^	0.362	0.373	<0.001	<0.001
Mesophilic bacteria	7.072	7.218	4.206 ^d^	6.559 ^c^	8.194 ^b^	9.619 ^a^	0.304	0.726	<0.001	0.031
Psychrotrophic bacteria	6.207	5.772	2.599 ^d^	6.006 ^c^	7.057 ^b^	8.294 ^a^	0.324	0.149	<0.001	0.008
Coagulase-positive staphylococci	5.552	4.916	3.316 ^b^	5.611 ^a^	6.242 ^a^	5.764 ^a^	0.263	0.439	<0.001	0.023
Coagulase negative staphylococci	5.087	4.885	3.065 ^c^	4.345 ^b^	6.045 ^a^	6.495 ^a^	0.406	0.698	<0.001	0.080
Lactic Acid Bacteria	4.059	4.197	1.846 ^c^	3.776 ^b^	5.275 ^a^	5.612 ^a^	0.277	0.887	<0.001	0.026
*Lactococcus* spp.	4.516	4.657	1.346 ^b^	5.567 ^a^	5.828 ^a^	5.601 ^a^	0.314	0.844	<0.001	0.570
Yeasts/Moulds	7.143	7.121	3.772 ^d^	6.203 ^c^	8.642 ^b^	9.908 ^a^	0.353	0.958	<0.001	0.037

^1^ CON, basal diet; AVI, basal diet + 0.5% organic acids and essential oils. ^2^ SEM, standard error of the means. ^3^
*p-Values* associated with dietary treatment (D), time of storage (T) and their interaction (D x T). ^4^ No significant differences were found for multiple comparisons using Tukey’s method. ^a, b, c, d^ Within row, different superscript letter indicates differences (*p <* 0.05) between times of storage tested using the Tukey’s Honest Significant Difference test.

**Table 5 animals-10-00640-t005:** Effect of the dietary treatments and time of storage on the oxidative stability of raw and cooked meat.

Item	Dietary Treatment (D) ^1^		Storage Time (T) ^2^				SEM^3^	*p*-Value ^4^		
	CON	AVI	0	1	2	3		D	T	D x T
*Raw meat*										
TBARS, mg/kg	0.88	0.84	0.76 ^c^	0.84 ^bc^	0.89 ^ab^	0.96 ^a^	0.015	0.441	<0.001	0.817
L* values	49.7	49.7	48.9 ^b^	49.1 ^b^	47.3 ^b^	53.5 ^a^	0.480	0.994	<0.001	0.874
a* values	2.97	3.59	2.35 ^b^	2.03 ^b^	2.55 ^b^	6.18 ^a^	0.210	0.075	<0.001	0.560
b* values	6.20	6.96	3.40 ^c^	4.42 ^bc^	5.76 ^b^	12.7 ^a^	0.439	0.284	<0.001	0.490
C* values	7.24	7.87	4.19 ^c^	4.89 ^c^	6.96 ^b^	14.2 ^a^	0.461	0.274	<0.001	0.190
H* values	64.8	61.4	52.8 ^b^	65.7 ^a^	69.8 ^a^	64.0 ^a^	0.141	0.141	<0.001	0.338
(K/S)_572_ ÷ (K/S)_525_	0.97	0.95	1.02 ^a^	1.00 ^a^	0.99 ^a^	0.82 ^b^	0.010	0.051	<0.001	0.428
ΔE ^5^ values	6.76	6.56	-	2.91 ^c^	4.97 ^b^	12.1 ^a^	0.577	0.806	<0.001	0.291
Whiteness index (WI)	48.9	48.9	48.7 ^ab^	48.8 ^ab^	46.8 ^b^	51.2 ^a^	0.408	0.941	<0.001	0.923
*Cooked meat*										
TBARS, mg/kg	5.10	4.00	1.20 ^d^	3.80 ^c^	5.60 ^b^	7.70 ^a^	0.310	0.004	<0.001	0.368
L* values	80.3	80.6	82.8 ^a^	79.5 ^b^	80.3 ^b^	79.3 ^b^	0.279	0.639	<0.001	0.214
a* values	4.45	4.47	3.83 ^b^	4.60 ^a^	4.42 ^ab^	4.99 ^a^	0.095	0.941	<0.001	0.049
b* values	12.9	12.5	10.9 ^c^	12.8 ^b^	12.9 ^b^	14.2 ^a^	0.167	0.311	<0.001	0.020
C* values	13.6	13.3	11.5 ^c^	13.7 ^b^	13.6 ^b^	15.1 ^a^	0.183	0.417	<0.001	0.019
H* values	71.0	70.6	70.7	70.5	71.2	70.8	0.242	0.526	0.674	0.257
(K/S)_572_ ÷ (K/S)_525_	1.11	1.07	0.96 ^b^	1.14 ^a^	1.11 ^a^	1.16 ^a^	0.017	0.224	<0.001	0.425
ΔE ^5^ values	4.60	4.47	-	4.22	4.07	5.32	0.284	0.868	0.047 ^6^	0.078
Whiteness index (WI)	76.0	76.4	79.2 ^a^	76.0 ^b^	75.3 ^b^	74.3 ^b^	0.300	0.535	<0.001	0.062

^1^ CON, basal diet; AVI, basal diet + 0.5% organic acids and essential oils. ^2^ Times 0, 1, 2, 3 correspond to: days 0, 4, 7, 11 (raw meat slices); days 0, 2, 4, 7 (cooked meat slices). ^3^ Standard error of the means (SEM). ^4^
*p-Values* associated with dietary treatment (D), time of storage (T) and their interaction (D x T). ^5^ Overall colour variation (ΔE) between each storage time and initial time. ^6^ No significant differences were found for multiple comparisons using Tukey’s method. ^a, b, c, d^ Within rows, different superscript letters indicate differences (*p* < 0.05) between times of storage or incubation, tested using the Tukey’s adjustment for multiple comparisons.

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
