# Peer review of "Effects of Microencapsulated Blend of Organic Acids and Essential Oils as a Feed Additive on Quality of Chicken Breast Meat"

_animals, 2020, doi:10.3390/ani10040640_

Round 1
Reviewer 1 Report
The paper presents some interesting results regarding the effect of microencapsulated blend of organics acids and essential oils on the shelf life of poultry meat. The research was performed with enough precision, however, some issues need to be clarified or added on the manuscript before it can be published:
According to the title, the ‘chicken breast/ m.pectoralis should be added in bracket. Please consider this.
Line 163: in which part of meat? In grounded or in the whole part of muscle?
L167: ‘Fifty’ should be in numeric notation. I suggest reformulating this sentence.
L179: What kind of tray it is - PP/PE, Styrofoam or different? What kind of cling film was used? Please report it.
L178: ‘Samples were split into four aliquots and stored at 4 °C on a tray covered 179 by a domestic cling film’. Samples of breast or samples of whole chicken?
L179-181: Authors stated that the meat shelf life of raw and cooked was measured in different days of storage. What is the reason of that choice and why the differences between storage are almost 4 days? Are there more studies using a similar method? In my opinion, to properly show and describe results is necessary to standardize the method and ‘days’. When analyzing test results, it is easy to make a mistake that the tests were carried out on the same days of storage, which lead to misunderstanding the whole results.
L207: How many times the measurement was performed? To the best of my knowledge, the average from 6 to 10 replicates is applied. The color measurement was carried on the whole or grounded meat, on which exact part of meat? It is well known that different parts of meat are varied.
L269 – 270, L281, L290 – this data can be added as a supplementary material. Please consider this.
L369-372: I cannot agree with this statement, however, it should be clearly described the differences between storage time of raw and cooked meat.
Discussion and material and method paragraph: please be consistent with the using meat samples and breast samples. It is misleading.
Author Response
The paper presents some interesting results regarding the effect of microencapsulated blend of organics acids and essential oils on the shelf life of poultry meat. The research was performed with enough precision, however, some issues need to be clarified or added on the manuscript before it can be published:
Response: Thank you very much for reviewing our manuscript.
According to the title, the ‘chicken breast/ m.pectoralis should be added in bracket. Please consider this.
Response: We thank the reviewer for the suggestion. The title has been changed into: “Effects of Microencapsulated Blend of Organics Acids and Essential Oils as a Feed Additive on Quality of Chicken Breast Meat“
Line 163: in which part of meat? In grounded or in the whole part of muscle?
Response: We used the whole part of meat. The sentence has been amended in the revised version to avoid confusion (L163-164).
L167: ‘Fifty’ should be in numeric notation. I suggest reformulating this sentence.
Response: We reformulated the sentence as suggested. Please check line 167
L179: What kind of tray it is - PP/PE, Styrofoam or different? What kind of cling film was used? Please report it.
Response: Both tray and film were in polyethylene. The missing information has been added (L178-179).
L178: ‘Samples were split into four aliquots and stored at 4 °C on a tray covered 179 by a domestic cling film’. Samples of breast or samples of whole chicken?
Response: We have added “breast” to avoid any misunderstanding (L178).
L179-181: Authors stated that the meat shelf life of raw and cooked was measured in different days of storage. What is the reason of that choice and why the differences between storage are almost 4 days? Are there more studies using a similar method? In my opinion, to properly show and describe results is necessary to standardize the method and ‘days’. When analyzing test results, it is easy to make a mistake that the tests were carried out on the same days of storage, which lead to misunderstanding the whole results.
Response: Several studies on shelf life of chicken meat consider a storage time of 14-15 days (or longer), with usually 3-4 sampling points. In the present work we established to evaluate the shelf life within 11 and 7 days of refrigerated storage (for raw and cooked meat, respectively) because longer periods do not reflect common domestic or retail conditions.
Please find below some studies that evaluated the shelf life in chicken meat (using conditions similar to ours):
- Luna, A., Labaque, M.C., Zygadlo, J.A. and Marin, R.H., 2010. Effects of thymol and carvacrol feed supplementation on lipid oxidation in broiler meat. Poultry Science, 89(2), pp.366-370. (days: 0, 5 and 10)
- Xue, S., Hu, J., Cheng, H. and Kim, Y.H.B., 2019. Effects of probiotic supplementation and postmortem storage condition on the oxidative stability of M. Pectoralis major of laying hens. Poultry science, 98(12), pp.7158-7169. (days: 1 and 7)
- Hernández-Hernández, E., Castillo-Hernández, G., González-Gutiérrez, C.J., Silva-Dávila, A.J., Gracida-Rodríguez, J.N., García-Almendárez, B.E., Di Pierro, P., Vázquez-Landaverde, P. and Regalado-González, C., 2019. Microbiological and Physicochemical Properties of Meat Coated with Microencapsulated Mexican Oregano (Lippia graveolens Kunth) and Basil (Ocimum basilicum L.) Essential Oils Mixture. Coatings, 9(7), p.414. (days: 0, 7, 14, 21 and 28)
L207: How many times the measurement was performed? To the best of my knowledge, the average from 6 to 10 replicates is applied. The color measurement was carried on the whole or grounded meat, on which exact part of meat? It is well known that different parts of meat are varied.
Response: As already specified in line 212, two readings were recorded for each sample. Which, in our opinion, are sufficient to have a representative measurement, considering the accuracy of the instrument and the homogeneity of the chicken breast color. As clearly reported in paragraph 2.2. (slaughter procedures and meat sampling), only the breasts were sampled at the slaughterhouse and transported to the laboratory for analysis. So, the color measurements were taken in duplicate on breast slices.
L269 – 270, L281, L290 – this data can be added as a supplementary material. Please consider this.
Response: We thank the reviewer for the suggestion. We have added a supplementary table as suggested.
L369-372: I cannot agree with this statement, however, it should be clearly described the differences between storage time of raw and cooked meat.
Response: We are sorry, but we do not understand what the reviewer refers to. This sentence simply reports what is already in the table 5. In other words, an effect of dietary treatment on lipid oxidation in cooked meat (P < 0.01) and the effect of storage time on TBARS, b* and C* values (P < 0.001).
Discussion and material and method paragraph: please be consistent with the using meat samples and breast samples. It is misleading.
Response: Thank you for bringing this to our attention. All the "breast samples" have been converted to "meat samples" in the revised version.
Reviewer 2 Report
The article concerns the effect of commercial feed additive on selected quality factors of chicken meat. The effect of so called label cleaning trend is affecting poultry manufacturers, therefore seeking for new better perceived feed additives is an important topic. The experiment is well designed and described. The manuscript is well written, organized and easy to read. Prior the publication consider following few minor suggestions:
Line 45-46 please rephrase
Line 95 please delete “as far we know,”
Line 209 please and CIE L*C*h color space after CIE L*a*b* (as it was presented in the study)
Figures 1 and 2 please describe Y axis in all figures
Table 5 please provide color difference ∆E between CON and AVI as well as between storage time to facilitate the reader, consider providing whiteness index. Both of these equations are calculated form CIE L*a*B* color space are described in Lewandowicz, J.; Le Thanh-Blicharz J. Quality of reduced fat mayonnaise prepared with native waxy starches. Proceedings of 14th International Conference on Polysaccharides-Glycoscience 2018, 262-265. Published online on Research Gate.
Line 476 please expand the statement regarding composition
Author Response
The article concerns the effect of commercial feed additive on selected quality factors of chicken meat. The effect of so called label cleaning trend is affecting poultry manufacturers, therefore seeking for new better perceived feed additives is an important topic. The experiment is well designed and described.
Response: Thank you very much for reviewing and appreciating our manuscript.
Line 45-46 please rephrase
Response: The sentence has been rephrased as requested (L44-46).
Line 95 please delete “as far we know,”
Response: Deleted as suggested (L95).
Line 209 please and CIE L*C*h color space after CIE L*a*b* (as it was presented in the study)
Response: We are sorry, but we do not totally understand the comment. Trying to interpret it, the “CIE L*C*h color space” is a system derived from “CIE L*a*b*”, as C* and h* values are calculated from a* and b* coordinates. Usually, only “CIE L*a*b*” system is reported in scientific articles.
Please note that even according to the official Meat Color Measurement Guidelines (AMSA, 2012), only the wording “CIE L*a* b*” is used and never “CIE L*C*h”.
Figures 1 and 2 please describe Y axis in all figures
Response: Described as requested.
Table 5 please provide color difference ∆E between CON and AVI as well as between storage time to facilitate the reader, consider providing whiteness index. Both of these equations are calculated form CIE L*a*B* color space are described in Lewandowicz, J.; Le Thanh-Blicharz J. Quality of reduced fat mayonnaise prepared with native waxy starches. Proceedings of 14th International Conference on Polysaccharides-Glycoscience 2018, 262-265. Published online on Research Gate.
Response: We thank the reviewer for the constructive comment. ∆E and WI (whiteness index) values were added in the table 5 as suggested.
Line 476 please expand the statement regarding composition
Response: This sentence has been modified in the revised version (L482-484).